# The Response of Weeds and Apple Trees to Beneficial Soil Microorganisms and Mineral Fertilizers Applied in Orchards

Jerzy Lisek *, Lidia Sas-Paszt, Augustyn Mika and Anna Lisek

The National Institute of Horticultural Research, Konstytucji 3 Maja 1/3 Str., 96-100 Skierniewice, Poland
* Correspondence: jerzy.lisek@inhort.pl

**Abstract:** The effect of beneficial soil fungi, bacteria and mineral fertilizers containing nitrogen, phosphorus and potassium on the level and species composition of weed infestation, uptake of macronutrients by weeds and the nutritional status of 'Sampion' apple trees with nitrogen (N), phosphorus (P), potassium (K), magnesium (Mg) and calcium (Ca) was assessed in three-year studies. In the field experiment, the effect of thirteen treatments was investigated, where fungal or bacterial inocula and mineral fertilizers at a standard dose and a dose reduced by 40% were applied individually or together. The fungal inoculum contained two species: *Aspergillus niger* and *Purpureocillium lilacinum*. The bacterial consortium consisted of three *Bacillus* strains: *Bacillus* sp., *Bacillus amyloliquefaciens* and *Paenibacillus polymyxa*. The weight of weeds and the uptake of macronutrients by weeds in the apple orchard increased significantly more after the application of mineral fertilization than after the application of beneficial organisms. The increased uptake of macronutrients by weeds did not significantly change the mineral nutritional status of apple trees expressed as the content of N, P, K Mg and Ca in apple leaves. After the use of NPK mineral fertilization, also with the addition of beneficial organisms, the uptake of N by both weeds and apple trees increased. P and K were more efficiently absorbed by weeds and they achieved greater benefits from fertilization with these two macroelements than trees.

**Keywords:** orchard; spontaneous vegetation; competition; macronutrients; nutrient uptake; nutritional status; bacteria; fungi

## 1. Introduction

A total of 86.4 million tons of apples are produced annually in the world [1]. Poland, with an annual production of 3–4 million tons, is among the most important producers of this fruit in the world, and apple orchards in Poland cover an area of over 170,000 hectares [2]. Apple trees require proper orchard floor management (OFM) to ensure good protection against weeds, availability of nutrients, good growth and yield of trees and high-quality fruit [3–9]. Weeds compete with cultivated plants for light, water, and nutrients, have an allelopathic effect and increase the risks caused by diseases, pests, including rodents and spring frosts during the flowering of fruit trees [10]. It should be emphasized that the presence of synanthropic flora, of which weeds form the basis, also brings benefits, referred to as ecosystem services, such as increasing biodiversity, creating a habitat for beneficial organisms, protecting the soil from erosion, salinity and mechanical compaction, reducing soil nutrient leaching, increasing the organic matter content in the soil, landscape and ornamental functions [10–12].

Weed–crop competition is conditioned by many factors, such as the species composition of weed communities, weed density (abundance), the ability to compete with the crop, and the availability of nutrients in the soil, and fertilization makes these relationships even more complex, as the question arises whether weeds or crops will benefit more from fertilization [13, 14]. Plant species, including weeds, differ in nutrient preferences [15–22]. Their preferences for nitrogen availability (nitrophily) are important and relatively well known [23,24]. Most of the research on competition is carried out in the single species weed annual crop system,

and the results indicate a diversified ability of crops, to compete for nutrients with individual species of weeds [25–27]. Contrary to the experimental model, weed communities containing several or more species are most often found in agricultural areas in arable lands [14,28].

Total nutrient uptake by weeds depends primarily on the nutrient content of the weeds and the weed biomass, which is limited by weed control treatments. South Korean persimmon orchard weeds contained N: 1.5–2.96; P: 0.23–0.41; K: 2.85–4.46% DW [29]. Vegetation can take up 41.4 kg N, 11.9 kg P, and 89.6 kg K per ha during the year in a pear orchard of South Korea [30]. Weeds growing in Polish orchards contained N: 2.39–3.60; P: 0.31–0.64; K: 1.15–3.35; Ca: 0.2–1.28; Mg: 0.12–0.23% DW; and the differences in the macronutrient content between species were significant [10]. After destruction, weeds undergo gradual mineralization, as a result of which the nutrients taken up by the weeds can be used by crops [29,31,32]. The most important role in this transformation is played by soil microorganisms and the plant rhizosphere [33,34].

In order to meet the food needs of the growing human population, conventional agriculture relies primarily on chemical fertilizers and pesticides [35]. The use of pesticides, fertilizers and often soil disturbance pose the greatest threats to soil functionality and biodiversity [36]. Soil-related biodiversity plays a fundamental role in the delivery of ecosystem services in terrestrial ecosystems, including agricultural land [37,38]. Bacteria and fungi as the essential components of this biodiversity are involved in basic ecological functions such as the circulation of nutrients and carbon sequestration [39,40], which affect the health of plants [41]. Increasing the role and use of the potential of beneficial soil microorganisms takes place in two ways, through the implementation of agricultural practices promoting microbial communities, which is better documented in agricultural crops than in perennial crops, such as orchards [42] and the use of the so-called biofertilizers [43]. Organic cultivation of an apple orchard increases the diversity of arbuscular fungi [44] and bacteria networks [45] compared to conventional cultivation.

Biofertilizers are products which contain living microorganisms, which, when applied to seed, plant surfaces, or soil, colonize the rhizosphere or the interior of the plant, and promote growth by increasing the supply or availability of primary nutrients to the host plant [46]. As an active ingredient of biofertilizers, they often use plant growth-promoting microorganisms (PGPMs), the best documented of which are arbuscular mycorrhizal fungi (AMF) [47] and plant growth-promoting rhizobacteria (PGPR) [48]. These groups of microorganisms play a useful role in plant nutrition, including the uptake of macronutrients such as N, P and K [49]. Atmospheric $N_2$ fixation is carried out by both bacteria symbiotic with Fabaceae plants, belonging to the family Rhizobiaceae (i.e., *Rhizobium*, *Bradyrhizobium*, *Azorhizobium*, *Mesorhizobium*, and *Sinorhizobium* (*Ensifer*)) [50], and by several heterotrophic free-living diazotrophic microorganisms such as *Azotobacter* sp., *Azospirillum* sp., and cyanobacteria [51]. Soil microorganisms belonging to genera *Pseudomonas*, *Bacillus*, *Rhizobium*, *Enterobacter*, *Penicillium*, and *Aspergillus* convert the insoluble forms of P into the form accessible to plants [52,53]. Soil bacteria (e.g., *Bacillus*, *Rhizobium*, *Acidithiobacillus*, *Paenibacillus*, *Pseudomonas*, and *Burkholderia*) and fungi (*Aspergillus*, *Cladosporium*, *Macrophomina*, *Sclerotinia*, *Trichoderma*, *Glomus*, and *Penicillium*) increase the availability of K to plants in the solubilization process [54,55]. The effect of biofertilizers in the field is related to many factors: plant related (cultural plant genotype and physiological status), edaphic/environmental (biotic, abiotic, and agricultural practices), and inoculant related (genotype, concentration, and formulation) [49,56,57]. Biofertilizers can positively affect the development, yield and mineral status of fruit trees of various species [58,59], including apple trees [60–63].

Plant-associated *Bacillus amyloliquefaciens* strains are increasingly used as biofertilizer and biocontrol agents in agriculture [64,65]. *B. amyloliquefaciens* can improve soil nutrient availability for plants, including improving nitrogen supply, solubilizing phosphate and potassium, and producing siderophores to enhance the availability of iron in the soil, can change the soil microbial community and improve the availability of minerals and plant growth conditions and can enhance plant resistance against biotic and abiotic stresses [66]. Root inoculation with *Bacillus* M3, *Bacillus* OSU-142 and *Microbacterium* FS01 strains (applied

alone or in combinations) significantly increased the yield, growth and leaf nutrient (nitrogen, phosphorus, potassium, calcium, iron, manganese, and zinc) contents in apples 'Granny Smith'/MM 106 [67].

*Purpureocillium lilacinum* (formerly *Paecilomyces lilacinus*) is a fungus endophytically colonizing plants and inoculation with its strains distinctly increased the availability of P and N and promoted the growth of maize, bean and soybean [68]. *Aspergillus niger* used together with another fungus, *Penicillium oxalicum*, had a positive influence on the biomass of maize plants by dissolving phosphorus compounds [69]. *Purpureocillium lilacinum* and *Aspergillus niger* are known as biocontrol agents—alone and in combination, they reduced infestation by pathogenic bacteria and nematodes on carrots [70]. Biofertilizers are primarily developed as an alternative to synthetic (mineral) fertilizers [43], but attempts are being made to use them together to improve the availability of synthetic fertilizers by plants and reduce their doses [71,72].

The aim of the experiment was to assess the effects of the use of mineral fertilizers and beneficial microorganisms (bacterial and fungal inocula), applied at different doses or in different ways, alone and in combination, on the nutritional status of apple trees and on the growth of weeds in field conditions. An important element of this assessment was to determine the uptake of macronutrients by the aboveground part of weeds and to clarify whether the placement of fertilizers and beneficial microorganisms (biofertilizers) can alter the competition of weeds with crops under periodic in-row herbicide fallow. The apple tree was selected for this study because it is the most important fruit crop in the temperate climate zone.

## 2. Materials and Methods

### 2.1. Experimental Site and Material

The field experiment was conducted in the Experimental Orchard of the National Research Institute of Horticulture in Dąbrowice, Central Poland (51°55′ N, 20°06′ E, 145 m a.s.l.), Central Europe. Orchard soil type was classified as luvisol, according to the international soil classification system [73]. Floatable parts (silt and clay), with a diameter less than 0.02 mm, marked according to Casagrande's method of aerometry, modified by Prószyński [74], constituted 15% of soil granulometric composition. At the time of planting, the pH of soil was 6.2 (in KCl), and the average organic matter content of the soil was 1.2%. The levels of available macroelements in the soil were as follows: phosphorus (P)—7.5, potassium (K)—12.4, and magnesium (Mg)—5.8 mg 100 g. The research was conducted in the years 2019–2021 on apple trees. The temperate climate of Central Poland, intermediate between maritime and continental, is characterized by cold winters and hot summers, and relatively low and changeable precipitation. In the years 2019–2021, the average air temperature was −0.4 °C in January (coldest month) and 19.7 °C in July (warmest month). The average air temperature and precipitation during growing season (April–October) are shown in Table 1.

**Table 1.** Average air temperature and total precipitation in Dąbrowice in the 2019–2021 growing seasons.

| Average Air Temperature (°C) Year/Month | Apr | May | Jun | Jul | Aug | Sep | Oct | Average Apr–Oct |
|---|---|---|---|---|---|---|---|---|
| 2019 | 9.6 | 12.5 | 21.7 | 18.3 | 19.8 | 13.9 | 10.0 | 15.1 |
| 2020 | 8.0 | 11.1 | 17.7 | 16.8 | 18.9 | 14.0 | 9.3 | 13.7 |
| 2021 | 6.7 | 12.5 | 19.9 | 21.2 | 16.8 | 13.7 | 9.3 | 14.3 |
| 1991–2020 * | 8.7 | 13.7 | 17.2 | 19.1 | 18.5 | 13.9 | 8.4 | 14.2 |
| **Total Precipitation (mm) Year/Month** | **Apr** | **May** | **Jun** | **Jul** | **Aug** | **Sep** | **Oct** | **Total Apr–Oct** |
| 2019 | 15.6 | 52.2 | 36.4 | 50.8 | 59.2 | 82.2 | 26.4 | 322.8 |
| 2020 | 9.6 | 81.6 | 135.0 | 63.2 | 110.4 | 74.8 | 99.2 | 573.8 |
| 2021 | 62.0 | 62.6 | 50.6 | 136.4 | 162.2 | 35.2 | 8.2 | 517.2 |
| 1991–2020 * | 40.7 | 60.3 | 71.7 | 75.3 | 55.6 | 50.1 | 43.3 | 397.0 |

* 30-year average.

## 2.2. Experimental Design

Treatments were applied in completely randomized blocks with 4 replications and 3 trees on the plots (12 trees per treatment). The width of the plots was 2 m and their area was 12 m$^2$.

From 2019 to 2021, the following methods of soil fertilization were introduced:

1.  Control—no fertilization.
2.  Beneficial fungi (fung.)—beneficial soil fungi on their own in the amount of 5.25 g per plot were applied. The mixture of beneficial soil fungi contained two species: *Aspergillus niger* and *Purpureocillium lilacinum*. The fungal spores concentration was approx. $10^8$ cfu g.
3.  Beneficial bacteria (bact.)—beneficial soil bacteria on their own in the amount of 3.83 g of overgrown substrate of concentration $1-2 \times 10^8$ cfu g per plot was applied. The mixture of beneficial bacteria contained three strains of *Bacillus* (*Bacillus* sp., *Bacillus amyloliquefaciens* and *Paenibacillus polymyxa*).
4.  Standard NPK soil fertilization. Mineral fertilizers at doses of 220 g of urea, 60 g of Super Fos Dar 40 granulated fertilizer, and 160 g of potassium salt were used per plot.
5.  Standard NPK soil fertilization as in point 4 with the beneficial soil fungi listed in point 2.
6.  Standard NPK soil fertilization as in point 4 with the beneficial bacteria listed in point 3.
7.  Polifoska 6 at a 100% dose—150 g per plot + NK (urea—192 g and K salt—85 g per plot).
8.  Polifoska 6 at a 100% dose as in point 7 enriched with beneficial bacteria listed in point 3 + NK (urea and K salt) as in point 7.
9.  Urea at a 100% dose—220 g per plot enriched with beneficial fungi listed in point 2 + PK (S. Fos Dar 40 + K salt) as in point 4.
10. Super Fos Dar 40 at a 100% dose—60 g plot enriched with beneficial bacteria listed in point 3 + NK (urea + K salt) as in point 4.
11. Urea in a 60% dose—132 g per plot enriched with beneficial fungi listed in point 2 fung. + 60% PK (S. Fos Dar 40–36 g + K salt—96 g per plot).
12. Polifoska 6 at a 60% dose—90 g per plot enriched with beneficial bacteria listed in point 3 + 60% NK (urea—115 g and K salt—51 g per plot).
13. Super Fos Dar 40 at a 60% dose—36 g per plot enriched with beneficial bacteria listed in point 3 + 60% NK (urea + K salt) as in point 12.

In the treatments marked as points 4–10, the total doses of nutrients amounted to: 80 kg N; 25 kg $P_2O_5$; 80 kg $K_2O$ per ha. In treatments 11–13, nutrients were used at total doses: 48 kg N; 15 kg $P_2O_5$; 48 kg $K_2O$ per ha (60% of standard doses). In 2018, mineral fertilizers and beneficial organisms were applied at doses as described, only the urea dose was $\frac{1}{4}$ of that used in 2019–2021. Fertilizers and beneficial microorganisms were applied in April. Mineral fertilizers were spread onto the soil surface. The microorganisms in treatments 2, 3, 5, 6 were mixed with the topsoil. In treatments 8–13, the microorganisms were mixed with urea or Polifoska 6 or Super Fosdar 40 fertilizers and spread onto moist soil during rainy weather. The numbering of the treatments (1–13) used in this subsection is also shown in appropriate tables containing the results.

## 2.3. Characteristics of the Fertilizers Used

Super Fos Dar 40 contains 40% $P_2O_5$—phosphorus pentoxide soluble in mineral acids; 25% $P_2O_5$ soluble in a neutral citrate solution and water; 10% CaO—calcium oxide soluble in water, and microelements (Co, Cu, Fe, Mn, and Zn), which are valuable additions, derived from natural phosphorites, that improve the assimilation of other ingredients. Polifoska 6 is a granulated fertilizer that contains 6% nitrogen (N) in the ammonium form, 20% phosphorus ($P_2O_5$), 30% potassium ($K_2O$) in the form of a potassium salt, and 7% sulfur trioxide ($SO_3$) soluble in water in the form of sulfate. Urea contains 46% nitrogen (N) in the amide form. Potassium salt contains 60% potassium ($K_2O$).

### 2.4. Measurements and Analyses

Every year in mid-June and mid-August, immediately before the control, weed infestation percentage of soil coverage weeds in total and by the most important weed species, fresh weight of the aboveground part of weeds hereinafter referred to as 'shoots', air-dry weight (air-DW) and absolute dry weight (DW) of weeds, macronutrient content in weed DW and uptake of macronutrients by weed shoots were measured. Seasonal fresh weight of weeds and seasonal macronutrient uptake by weeds were the sum of the measurements from June and August. In order to determine the air-DW of weeds, the samples of the plant material were dried at 60 °C with forced air circulation. To determine the absolute dry weight, the air-dried sample was dried at 105 °C for at least 3 h until a constant weight was obtained. Macronutrient uptake was calculated by multiplying the DW of weeds taken per ha by the content of macronutrients in DW. In mid-August of each year, the content of macronutrients in apple leaves was determined. Leaves with petioles were picked from the middle part of non-bearing one-year-old shoots, from various parts of tree canopies, at the height of 1.5–2 m. P, K and Mg concentration in soil was analyzed in mid August 2021, in the last year of research. Soil samples were taken at a depth of 0–30 cm, assuming that soil fertilization will primarily affect its surface layer. Macroelement concentration in leaf tissue and soil was analyzed at the accredited Chemical Pollution Research Laboratory of the National Research Institute of Horticulture, Skierniewice, Poland. For the determination of available forms of P and K in a mineral soil, the Egner-Riehm method was used [75]. The method consists of extracting P and K compounds from the soil by means of calcium lactate. For the determination of available forms of Mg in a mineral soil, the Schachtschabel method was used [75]. The Schachtschabel method involves shaking soil with 0.025 M calcium chloride. The mineral content in plant material was determined through the process of mineralization (combustion) (VELP DK 42, VELP Scientifica Srl, Usmate, Italy). Wet combustion of dry, milled plant material consists of complete oxidation with liquid oxidants mixture: concentrated sulfuric acid (15%), nitric acid (75%) and perchloric acid (10%). For the determination of mineral concentration in the solutions obtained by the analytical methods mentioned above, measurements were carried out using the technique of atomic emission spectrometry with excitation in inductively coupled plasma (ICP-OES) (Spectrometer iCAP 6500 duo, Thermo Fisher Scientific, Waltham, Massachusetts, USA). DW was determined with the use of the weight method. Total N concentration in plant material was determined by the Dumas method (conductometric technique) using a TruSpec CNS analyzer (LECO Corporation, Saint Joseph, Michigan, USA). Nutritional status of bearing fruit trees was assessed by comparing N, P, K and Mg leaf concentration obtained through chemical analyses with Polish compositional standards [76]. Polish guidelines do not provide for the compositional standard of leaf Ca since its deficiency is rarely reported. Australian standards were used to assess the concentration of this macronutrient [77].

### 2.5. Statistical Analysis

Results were analyzed statistically using analysis of variance. The significance of the means was evaluated using Duncan's test at 5% level. Data are expressed as a percentage, concerning soil coverage with weeds, and air-DW content and macronutrient content were transformed according to the Bliss function. The correlation between macronutrient uptake by weed shoots and fresh weight of weed shoots, NPK fertilization level (0, 0.6, 1), beneficial microorganism level (0 or 1) and correlation between macronutrient content in apple leaves or macronutrients final content in the orchard soil and macronutrient uptake by weeds shoots, NPK fertilization level (0, 0.6, 1), beneficial microorganism level (0 or 1), was assessed using Pearson's linear correlation coefficient ($r$). The strength of Pearson's linear correlation coefficient was rated as follows: $|r|$: 0.0–0.2—very weak; 0.2–0.4—weak; 0.4–0.6—moderate; 0.6–0.8—strong; 0.8–1—very strong.

### 3. Results

Results for parameters such as soil coverage with weeds, air-DW content of weeds, and macronutrient content in apple leaves are only represented as three-year means. Data from individual years would not change the interpretation of the results. The variability of the above-mentioned parameters can be concluded on the basis of the standard deviation available in the appropriate tables.

### 3.1. Coverage of the Soil with Weeds

The most important weeds that covered at least 1% of the plot area in each of the years of research were *Poa annua* L. (POAAN), *Stellaria media* (L.) Vill. (STEME), *Capsella bursa-pastoris* (L.) Medik (CAPBP), *Senecio vulgaris* L. (SENVU), *Chenopodium album* L. (CHEAL), *Echinochloa crus-galli* (L.) Beauv. (ECHCG), *Amaranthus retroflexus* L. (AMARE) and *Lamium amplexicaule* L. (LAMAM). The mean total weed infestation, expressed as soil coverage with weeds, in all plots with mineral fertilization at a dose of 100 or 60%, where it amounted to 79.9–89.3%, was significantly higher than in plots where no mineral fertilizers were applied and reached the values of 64.9–65.2% (Table 2).

**Table 2.** Mean soil coverage with weeds in the apple orchard, 2019–2021.

| | Treatment | Soil Coverage (%) | | | | |
| --- | --- | --- | --- | --- | --- | --- |
| | | Total Weeds | AMARE * | CAPBP | CHEAL | ECHCG |
| 1. | No fertilization | 65.1 ± 24.8 a | 1.10 ± 0.40 a | 5.19 ± 1.37 a | 4.47 ± 2.44 a | 1.86 ± 1.50 a |
| 2. | Fungal strains (fung.) to soil | 64.9 ± 25.9 a | 1.22 ± 0.58 ab | 4.72 ± 1.48 a | 4.04 ± 2.55 a | 1.91 ± 1.18 a |
| 3. | Bacterial strains (bact.) to soil | 65.2 ± 24.1 a | 1.31 ± 0.77 a–c | 5.25 ± 1.15 a | 4.33 ± 2.39 a | 2.01 ± 1.20 ab |
| 4. | Standard NPK: urea + S. Fos Dar + potassium (K) salt | 88.4 ± 22.6 b | 2.03 ± 0.58 d–f | 8.61 ± 1.68 b–d | 7.17 ± 2.60 b | 2.97 ± 1.87 d |
| 5. | Standard NPK + fung. to soil | 87.8 ± 21.9 b | 2.09 ± 0.54 ef | 8.37 ± 2.15 b–d | 6.89 ± 3.79 b | 2.94 ± 1.45 d |
| 6. | Standard NPK + bacter. to soil | 89.3 ± 23.5 b | 2.00 ± 0.40 d–f | 8.94 ± 2.33 cd | 7.39 ± 4.10 b | 2.95 ± 1.50 d |
| 7. | 100% Polifoska 6 + NK (urea + K salt) | 88.5 ± 18.5 b | 1.78 ± 0.82 c–f | 9.06 ± 2.64 cd | 7.23 ± 3.18 b | 2.75 ± 1.46 d |
| 8. | 100% Polifoska 6 en. bacter. + NK (urea + K salt) | 89.1 ± 17.0 b | 1.98 ± 0.58 d–f | 9.66 ± 1.95 d | 6.88 ± 2.80 b | 2.64 ± 1.42 cd |
| 9. | 100% urea en. fung. + PK (S. Fos Dar + K salt) | 88.1 ± 21.7 b | 2.11 ± 0.78 f | 9.32 ± 2.51 cd | 7.08 ± 2.18 b | 2.97 ± 1.65 d |
| 10. | 100% S. Fos Dar en. bact.+ NK (urea + K salt) | 79.9 ± 21.5 b | 1.52 ± 0.73 b–d | 7.80 ± 1.50 bc | 6.62 ± 2.33 b | 3.01 ± 1.68 d |
| 11. | 60% urea en. fung. + 60% PK (S. Fos Dar + K salt) | 85.1 ± 26.3 b | 1.57 ± 0.83 b–e | 7.70 ± 0.94 bc | 6.32 ± 1.34 b | 2.54 ± 1.31 b–d |
| 12. | 60% Polifoska 6 en. bacter. + 60% NK (urea + K salt) | 82.8 ± 22.1 b | 1.53 ± 0.78 b–d | 7.22 ± 1.48 b | 6.17 ± 1.95 b | 2.18 ± 1.10 a–c |
| 13. | 60% S. Fos Dar en. bacter. + 60% NK (urea + K salt) | 80.0 ± 20.7 b | 1.67 ± 0.80 b–f | 7.55 ± 1.36 bc | 6.10 ± 2.25 b | 2.09 ± 1.03 a–c |

Means followed by the same letter do not differ significantly at $p = 0.05$. Values with the prefix ± represent the standard deviation. * Weed names according to the Bayer code: AMARE—*Amaranthus retroflexus*, CAPBP—*Capsella bursa-pastoris*, CHEAL—*Chenopodium album*, and ECHCG—*Echinochloa crus-galli*.

The soil coverage with weeds belonging to species such as CAPBP, CHEAL, SENVU and STEME was significantly greater in plots with mineral fertilization, regardless of its level, than in plots where such fertilization was not applied (Tables 2 and 3).

**Table 3.** Mean soil coverage with weeds in the apple orchard, 2019–2021.

| Treatment | Soil Coverage (%) | | | |
|---|---|---|---|---|
| | LAMAM * | POAAN | SENVU | STEME |
| 1. No fertilization | 1.12 ± 0.67 a | 29.3 ± 9.27 a | 5.19 ± 1.56 a | 7.42 ± 1.68 a |
| 2. Fungal strains (fung.) to soil | 1.11 ± 0.64 a | 30.0 ± 9.61 ab | 4.45 ± 1.55 a | 7.40 ± 1.15 a |
| 3. Bacterial strains (bact.) to soil | 1.25 ± 0.70 a | 30.0 ± 11.7 ab | 4.99 ± 1.10 a | 8.19 ± 1.71 a |
| 4. Standard NPK: urea + S. Fos Dar + potassium (K) salt | 1.49 ± 0.85 a | 36.6 ± 12.2 a–c | 6.82 ± 1.50 b | 10.5 ± 1.79 bc |
| 5. Standard NPK + fung. to soil | 1.61 ± 0.87 a | 37.8 ± 12.8 bc | 6.58 ± 1.60 b | 10.4 ± 1.53 b |
| 6. Standard NPK + bacter. to soil | 1.56 ± 0.90 a | 38.6 ± 10.3 c | 6.52 ± 1.38 b | 10.2 ± 1.43 b |
| 7. 100% Polifoska 6 + NK (urea + K salt) | 1.48 ± 0.81 a | 36.0 ± 10.3 a–c | 7.04 ± 1.87 b | 12.8 ± 1.93 d |
| 8. 100% Polifoska 6 en. bacter. + NK (urea + K salt) | 1.56 ± 0.83 a | 37.2 ± 10.8 a–c | 7.27 ± 1.23 b | 12.3 ± 1.69 cd |
| 9. 100% urea en. fung. + PK (S. Fos Dar + K salt) | 1.58 ± 0.85 a | 35.8 ± 9.8 a–c | 7.13 ± 1.68 b | 11.9 ± 1.97 b–d |
| 10. 100% S. Fos Dar en. bact.+ NK (urea + K salt) | 1.52 ± 0.82 a | 34.5 ± 10.5 a–c | 6.80 ± 1.77 b | 11.7 ± 1.56 b–d |
| 11. 60% urea en. fung. + 60% PK (S. Fos Dar + K salt) | 1.36 ± 0.72 a | 36.2 ± 12.8 a–c | 6.44 ± 1.60 b | 10.9 ± 1.74 b–d |
| 12. 60% Polifoska 6 en. bacter. + 60% NK (urea + K salt) | 1.21 ± 0.69 a | 37.6 ± 11.5 a–c | 6.58 ± 1.72 b | 10.5 ± 1.33 bc |
| 13. 60% S. Fos Dar en. bacter. + 60% NK (urea + K salt) | 1.10 ± 0.65 a | 35.1 ± 11.9 a–c | 6.58 ± 1.70 b | 10.1 ± 1.24 b |

Means followed by the same letter do not differ significantly at $p$ = 0.05. Values with the prefix ± represent the standard deviation. * Weed names according to the Bayer code: LAMAM—*Lamium amplexicaule,* POAAN—*Poa annua*, SENVU—*Senecio vulgaris*, and STEME—*Stellaria media*.

The soil coverage of AMARE weeds was significantly greater in all plots with mineral fertilizers compared to the untreated control plots. The response of ECHCG cover to fertilization, was visible mainly after the application of higher doses of mineral fertilizers. POAN showed a slight reaction to mineral fertilization and the coverage of the soil with plants of this species on plots with the NPK standard, combined with the use of bacterial or fungal strains mixed with the soil, was significantly higher only compared to the untreated plots. The only species that did not respond to fertilization was LAMAM. The use of fungal or bacterial strains did not change the level of soil cover with any species of weed compared to the untreated check. Coverage of all species in plots with NPK standard and NPK standard + fungal strains or bacterial strains was the same. Additionally, the use of Polifoska 6 + NK (urea + K salt) at a dose of 100% enriched with bacterial strains did not change the occurrence of any of the weed species compared to 100% Polifoska 6 + NK (urea + K salt) applied alone.

*3.2. Fresh Weight of Weed Shoots*

There were differences between treatments regarding this parameter in individual seasons. Mean seasonal fresh weight of weed shoots from three years (2019–2021), collected from plots where mineral fertilizers were applied at the full dose, except for treatment 100% S. Fos Dar en. bact. + NK (urea + K salt), amounting to 1041–1174 g m$^2$, was significantly higher than in the plots of three treatments, where no mineral fertilizers were applied, and its value was 766–804 g m$^2$ (Table 4).

**Table 4.** Fresh weight of weeds and air-DW content of weeds in the apple orchard, 2019–2021.

| Treatment | Fresh Weight of Weeds (g m) | | | | Air-DW Content of Weeds (%) |
|---|---|---|---|---|---|
| | **2019** | **2020** | **2021** | **Mean for 2019–2021** | **Mean for 2019–2021** |
| 1. No fertilization | 905 ± 62.4 ab | 740 ± 81.5 ab | 664 ± 49.4 a | 770 ± 123 a | 15.4 ± 2.78 ab |
| 2. Fungal strains (fung.) to soil | 839 ± 55.6 a | 717 ± 45.0 a | 741 ± 49.3 b | 766 ± 65 a | 15.6 ± 2.61 ab |
| 3. Bacterial strains (bact.) to soil | 954 ± 85.9 b | 722 ± 65.9 ab | 735 ± 32.4 b | 804 ± 130 a | 15.5 ± 2.93 ab |
| 4. Standard NPK: urea + S. Fos Dar + potassium (K) salt | 1457 ± 82.7 de | 1228 ± 91.5 g | 837 ± 107.6 cd | 1174 ± 314 b | 14.1 ± 2.08 a |
| 5. Standard NPK + fung. to soil | 1486 ± 33.7 e | 1082 ± 5.6 ef | 844 ± 40.7 cd | 1137 ± 325 b | 15.1 ± 2.28 ab |
| 6. Standard NPK + bacter. to soil | 1345 ± 97.4 cd | 986 ± 66.3 d | 901 ± 62.4 de | 1077 ± 234 b | 15.3 ± 2.33 ab |
| 7. 100% Polifoska 6 + NK (urea + K salt) | 1440 ± 92.3 de | 1036 ± 62.2 de | 842 ± 15.7 cd | 1106 ± 305 b | 14.8 ± 2.85 ab |
| 8. 100% Polifoska 6 en. bacter. + NK (urea + K salt) | 1472 ± 40.6 e | 1120 ± 54.3 f | 851 ± 55.7 c–e | 1148 ± 311 b | 14.5 ± 1.74 ab |
| 9. 100% Urea en. fung. + PK (S. Fos Dar + K salt) | 1307 ± 59.4 c | 896 ± 49.9 c | 921 ± 20.8 e | 1041 ± 230 b | 15.0 ± 2.33 ab |
| 10. 100% S. Fos Dar en. bact.+ NK (urea + K salt) | 1300 ± 99.6 c | 804 ± 42.1 b | 791 ± 14.4 bc | 965 ± 290 ab | 15.0 ± 1.80 ab |
| 11. 60% urea 60% en. fung. + 60% PK (S. Fos Dar + K salt) | 1470 ± 129 e | 699 ± 27.5 a | 802 ± 12.4 bc | 990 ± 418 ab | 14.6 ± 1.27 ab |
| 12. 60% Polifoska 60% en. bacter. + 60% NK (urea + K salt) | 1379 ± 31.3 c–e | 730 ± 36.8 ab | 742 ± 27.4 b | 950 ± 371 ab | 15.7 ± 1.96 b |
| 13. 60% S. Fos Dar 40 en. bacter. + 60% NK (urea + K salt) | 1289 ± 93.4 c | 885 ± 22.2 c | 732 ± 39.3 b | 969 ± 288 ab | 14.7 ± 1.70 ab |

Means followed by the same letter do not differ significantly at $p = 0.05$. Values with the prefix ± represent the standard deviation.

In 2019, the fresh weight of weed shoots on plots of all treatments with mineral fertilization was significantly higher than on plots without mineral fertilization. In 2020, fresh weed weight in control and plots treated with bacterial strains did not differ significantly compared to the three treatments: 100% S. Fos Dar en. bact. + NK (urea + K salt), 60% urea en. fung. + 60% PK (S. Fos Dar + K salt), 60% Polifoska 6 en. bacter. + 60% NK (urea + K salt). In 2021, fresh weed weight on plots with fungal or bacterial strains did not differ significantly compared to the four treatments: 100% S. Fos Dar en. bact. + NK (urea + K salt), 60% urea en. fung. + 60% PK (S. Fos Dar + K salt), 60% Polifoska 6 en. bacter. + 60% NK (urea + K salt), 60% and S. Fos Dar 60% en. bacter. + 60% NK (urea + K salt). The fresh weight of weed shoots on the plots without mineral fertilization, expressed as the mean for three years and in 2020, did not differ significantly between the three treatments. In 2019, the value of this parameter on plots treated with bacterial strains was significantly greater than after the use of fungal strains, while in 2021, the weight of weed shoots on plots with fungal or bacterial strains was significantly greater than on untreated plots. The fresh weight of weed shoots from treatments: standard NPK, standard NPK + fungi, standard NPK + bacteria did not differ significantly as the mean from 2019 to 2021 and in 2021, while in the 2019 and 2020 seasons, there were statistically significant differences in the value of this parameter. However, no significant differences in the fresh weight of weed shoots were found between the plots, where Polifoska 6 + NK (urea + K salt) at the full dose used alone or enriched with bacterial strains was applied.

### 3.3. The Content of Air-DW of Weed Shoots

The average value of this parameter over the three years was aligned. Only on the plots treated with 60% Polifoska 6 enriched with bacteria + 60% NK (urea + K salt), where

it amounted to 15.7%, was it significantly higher than on the plots fertilized with the NPK standard, where it amounted to 14.1% (Table 4).

### 3.4. Macronutrient Content in Weed Shoots

The mean N content in weed shoots in 2019–2021 was the highest in plots fertilized with 100% urea enriched with fungal strains + PK (S. Fos Dar + K salt) and amounted to 3.33% DW (Table 5).

**Table 5.** Mean macronutrient content in weed shoots in the apple orchard, 2019–2021.

| | Treatment | Mean Nutrient Content (%) | | | | |
|---|---|---|---|---|---|---|
| | | N | P | K | Mg | Ca |
| 1. | No fertilization | 2.78 ± 0.99 ab | 0.52 ± 0.09 a | 3.88 ± 0.63 a-c | 0.34 ± 0.10 a | 1.21 ± 0.34 a–c |
| 2. | Fungal strains (fung.) to soil | 2.68 ± 1.00 a | 0.55 ± 0.16 a | 3.73 ± 0.81 a | 0.34 ± 0.14 a | 1.20 ± 0.34 a–c |
| 3. | Bacterial strains (bact.) to soil | 2.73 ± 0.91 ab | 0.50 ± 0.09 a | 3.78 ± 0.55 ab | 0.33 ± 0.10 a | 1.25 ± 0.34 bc |
| 4. | Standard NPK: urea + S. Fos Dar + potassium (K) salt | 3.21 ± 0.58 bc | 0.52 ± 0.06 a | 4.59 ± 0.41 d | 0.34 ± 0.05 a | 1.28 ± 0.33 bc |
| 5. | Standard NPK + fung. to soil | 3.20 ± 0.58 bc | 0.47 ± 0.03 a | 4.11 ± 0.39 a–d | 0.31 ± 0.04 a | 1.33 ± 0.34 c |
| 6. | Standard NPK + bacter. to soil | 3.22 ± 0.81 bc | 0.53 ± 0.08 a | 4.40 ± 0.32 cd | 0.35 ± 0.09 a | 1.17 ± 0.23 a–c |
| 7. | 100% Polifoska 6 + NK (urea + K salt) | 3.19 ± 0.64 bc | 0.52 ± 0.07 a | 4.35 ± 0.37 b-d | 0.36 ± 0.04 a | 1.25 ± 0.34 bc |
| 8. | 100% Polifoska 6 en. bacter. + NK (urea + K salt) | 3.24 ± 0.55 bc | 0.52 ± 0.06 a | 4.47 ± 0.56 cd | 0.35 ± 0.04 a | 1.25 ± 0.28 bc |
| 9. | 100% Urea en. fung. + PK (S. Fos Dar + K salt) | 3.33 ± 0.81 c | 0.50 ± 0.07 a | 4.31 ± 0.81 b–d | 0.33 ± 0.09 a | 1.25 ± 0.26 bc |
| 10. | 100% S. Fos Dar en. bact.+ NK (urea + K salt) | 3.26 ± 0.77 bc | 0.49 ± 0.07 a | 4.39 ± 0.31 cd | 0.36 ± 0.06 a | 1.09 ± 0.20 ab |
| 11. | 60% urea 60% en. fung. + 60% PK (S. Fos Dar + K salt) | 3.22 ± 0.91 bc | 0.51 ± 0.10 a | 3.93 ± 0.62 a–c | 0.30 ± 0.07 a | 1.19 ± 0.36 a–c |
| 12. | 60% Polifoska 60% en. bacter. + 60% NK (urea + K salt) | 3.17 ± 0.64 a–c | 0.48 ± 0.05 a | 3.90 ± 0.46 a–c | 0.32 ± 0.04 a | 1.20 ± 0.31 a–c |
| 13. | 60% S. Fos Dar 40 en. bacter. + 60% NK (urea + K salt) | 3.08 ± 0.65 a–c | 0.49 ± 0.06 a | 4.17 ± 0.38 a–d | 0.31 ± 0.05 a | 1.01 ± 0.09 a |

Means followed by the same letter do not differ significantly at $p = 0.05$. Values with the prefix ± represent the standard deviation.

N content in weed shoots from plots treated with fungal strains (2.68% DW) was significantly lower than in all plots treated with mineral fertilizers in full dose and in plots with 60% urea en. fung. + 60% PK (S. Fos Dar + K salt). The content of P (0.47–0.55% DW) and Mg (0.31–0.36% DW) in weed shoots did not differ significantly between the treatments. The highest K content was found in the plots with the NPK standard (4.59% DW). K content in weed shoots from plots treated with fungal strains (3.73% DW) was significantly lower than in plots with 100% Polifoska 6 + NK (urea + K salt), 100% Polifoska 6 enriched with bacteria + NK (urea + K salt), 100% urea enriched fungi + PK (S. Fos Dar + K salt), 100% S. Fos Dar enriched bacteria + NK (urea + K salt). The highest Ca content was characteristic for weeds from plots treated with the NPK standard + fungi mixed with soil (1.33% DW), and the lowest (1.01% DW) from the treatment of 60% S. Fos Dar 40 enriched with bacteria + 60% NK (urea + K salt). There were no significant differences in the content of all macronutrients within treatments without mineral fertilization, within three treatments with the NPK standard (fertilizers alone or together with fungi or bacteria), and within a pair of 100% Polifoska 6 + NK (urea + K salt) and 100% Polifoska 6 enriched with bacteria + NK (urea + K salt).

### 3.5. Uptake of Macronutrients by Weed Shoots

On average, in the years 2019–2021, on plots where mineral fertilizers were applied at full dose, except for treatment S. Fos Dar 100% en. bact. + NK (urea + K salt), weed shoots took 49.3–53.5 kg N per ha between May and August, which was significantly more than 32.7–34.2 kg N per ha, taken from the plots of three treatments where no mineral fertilizers were applied (Table 6).

**Table 6.** N uptake by weed shoots in the apple orchard, 2019–2021.

| | Treatment | N Uptake (kg ha) | | | |
|---|---|---|---|---|---|
| | | **2019** | **2020** | **2021** | **Mean 2019–2021** |
| 1. | No fertilization | 29.9 ± 1.90 ab | 40.0 ± 3.92 ab | 28.4 ± 2.12 a | 32.8 ± 6.31 a |
| 2. | Fungal strains (fung.) to soil | 26.3 ± 2.13 a | 39.5 ± 2.65 ab | 32.4 ± 2.16 bc | 32.7 ± 6.61 a |
| 3. | Bacterial strains (bact.) to soil | 32.5 ± 2.72 b | 38.0 ± 3.80 a | 32.0 ± 1.42 bc | 34.2 ± 3.33 a |
| 4. | Standard NPK: urea + S. Fos Dar + potassium (K) salt | 60.7 ± 3.96 de | 62.3 ± 2.05 h | 34.4 ± 4.43 c–e | 52.5 ± 15.7 b |
| 5. | Standard NPK + fung. to soil | 62.7 ± 1.37 e | 58.1 ± 0.74 f–h | 39.6 ± 1.91 fg | 53.5 ± 12.2 b |
| 6. | Standard NPK + bacter. to soil | 57.2 ± 4.11 cd | 57.3 ± 4.41 e–g | 37.4 ± 2.57 e–g | 50.6 ± 11.5 b |
| 7. | 100% Polifoska 6 + NK (urea + K salt) | 62.1 ± 3.93 de | 54.0 ± 3.55 ef | 36.6 ± 0.69 d–f | 50.9 ± 13.0 b |
| 8. | 100% Polifoska 6 en. bacter. + NK (urea + K salt) | 62.0 ± 1.73 de | 59.3 ± 3.34 gh | 37.5 ± 2.44 e–g | 52.9 ± 13.4 b |
| 9. | 100% Urea en. fung. + PK (S. Fos Dar + K salt) | 54.4 ± 2.60 c | 53.4 ± 2.53 e | 40.0 ± 0.89 g | 49.3 ± 8.04 b |
| 10. | 100% S. Fos Dar en. bact.+ NK (urea + K salt) | 57.1 ± 4.37 cd | 47.2 ± 2.11 d | 29.4 ± 0.53 ab | 44.6 ± 14.0 ab |
| 11. | 60% urea 60% en. fung. + 60% PK (S. Fos Dar + K salt) | 62.2 ± 5.45 de | 45.0 ± 1.94 cd | 29.5 ± 0.36 ab | 45.6 ± 16.4 ab |
| 12. | 60% Polifoska 60% en. bacter. + 60% NK (urea + K salt) | 59.8 ± 1.38 de | 42.6 ± 1.39 bc | 34.0 ± 1.28 cd | 45.5 ± 13.1 ab |
| 13. | 60% S. Fos Dar 40 en. bacter. + 60% NK (urea + K salt) | 54.3 ± 4.20 c | 43.6 ± 2.18 b–d | 32.7 ± 1.75 c | 43.5 ± 10.8 ab |

Means followed by the same letter do not differ significantly at $p = 0.05$. Values with the prefix ± represent the standard deviation.

In 2019, weed shoots in all plots with mineral fertilization (full or reduced dose and treatments with beneficial organisms) took up significantly more N compared to the three treatments without mineral fertilization. In 2020 and 2021, there was a visible tendency for weeds to uptake more N in the fertilized plots with a full dose of mineral fertilizers, regardless of the use of beneficial microorganisms.

Mean P uptake by weed shoots in 2019–2021 was the highest in plots with the NPK standard + bacteria mixed with soil (8.68 kg ha) and it was significantly higher than in plots without mineral fertilization (6.37–6.78 kg ha) (Table 7).

The mean P uptake by weed shoots in 2019–2021 did not differ between the treatments without mineral fertilization, but already in individual years, the differences between these treatments were significant. In 2019, the uptake of P in the plots treated with fungal strains was significantly lower than in the control (no treatment) plots and treated with bacterial strains, while in 2020 and 2021 it was significantly higher than in these two treatments. The addition of bacteria to Polifoska 6 + NK (urea + K salt) at the full dose resulted in a higher P uptake compared to 100% Polifoska 6 + NK (urea + K salt) only in 2020. Differences between the treatments of standard NPK, standard NPK + fungal strains to soil, standard NPK + bacterial strains to soil in the uptake of P by weed shoots, calculated as a three-year mean and in 2019, were insignificant. In 2020, the uptake of P by weeds on the NPK standard plots was significantly higher than on plots with the NPK standard + fungi and the NPK

standard + bacteria. In 2021, the P uptake of the NPK standard + bacteria treated plots was significantly higher than on the plots treated with NPK standard and NPK standard + fungi.

**Table 7.** P uptake by weed shoots in the apple orchard, 2019–2021.

| | Treatment | P Uptake (kg ha) | | | |
|---|---|---|---|---|---|
| | | **2019** | **2020** | **2021** | **Mean 2019–2021** |
| 1. | No fertilization | 6.38 ± 0.41 b | 6.05 ± 0.56 ab | 6.68 ± 0.50 ab | 6.37 ± 0.32 a |
| 2. | Fungal strains (fung.) to soil | 5.32 ± 0.40 a | 6.99 ± 0.43 cd | 8.03 ± 0.53 ef | 6.78 ± 1.37 a–c |
| 3. | Bacterial strains (bact.) to soil | 6.11 ± 0.52 b | 5.99 ± 0.60 ab | 7.46 ± 0.33 c–e | 6.52 ± 0.82 ab |
| 4. | Standard NPK: urea + S. Fos Dar + potassium (K) salt | 9.46 ± 0.57 ef | 9.58 ± 0.59 h | 6.60 ± 0.85 a | 8.55 ± 1.69 b–d |
| 5. | Standard NPK + fung. to soil | 9.34 ± 0.21 d–f | 7.71 ± 0.27 ef | 7.10 ± 0.34 a–c | 8.05 ± 1.16 a–d |
| 6. | Standard NPK + bacter. to soil | 9.42 ± 0.67 ef | 8.20 ± 0.62 f | 8.41 ± 0.58 f | 8.68 ± 0.65 d |
| 7. | 100% Polifoska 6 + NK (urea + K salt) | 9.24 ± 0.57 d–f | 7.53 ± 0.57 de | 7.79 ± 0.15 d–f | 8.19 ± 0.92 a–d |
| 8. | 100% Polifoska 6 en. bacter. + NK (urea + K salt) | 9.63 ± 0.29 f | 8.90 ± 0.48 g | 7.16 ± 0.47 a–d | 8.56 ± 1.27 cd |
| 9. | 100% Urea en. fung. + PK (S. Fos Dar + K salt) | 9.09 ± 0.46 d–f | 6.50 ± 0.21 bc | 7.81 ± 0.18 d–f | 7.80 ± 1.30 a–d |
| 10. | 100% S. Fos Dar en. bact.+ NK (urea + K salt) | 8.29 ± 0.63 c | 5.96 ± 0.32 b | 7.09 ± 0.13 a–c | 7.11 ± 1.17 a–d |
| 11. | 60% urea 60% en. fung. + 60% PK (S. Fos Dar + K salt) | 9.70 ± 0.85 f | 6.09 ± 0.26 ab | 7.35 ± 0.12 b–d | 7.71 ± 1.83 a–d |
| 12. | 60% Polifoska 60% en. bacter. + 60% NK (urea + K salt) | 8.81 ± 0.20 c–e | 5.63 ± 0.31 a | 6.89 ± 0.26 a–c | 7.11 ± 1.60 a–d |
| 13. | 60% S. Fos Dar 40 en. bacter. + 60% NK (urea + K salt) | 8.59 ± 0.63 cd | 5.96 ± 0.09 ab | 6.89 ± 0.37 a–c | 7.15 ± 1.33 a–d |

Means followed by the same letter do not differ significantly at $p$ = 0.05. Values with the prefix ± represent the standard deviation.

K uptake, expressed as the mean from 2019 to 2021, in all plots with the application of full-dose mineral fertilizers reached 62.5–75.7 kg ha and was significantly higher or showed a clear tendency to higher uptake, compared to the treatments without mineral fertilization, where it was 45.0–48.7 kg ha (Table 8).

**Table 8.** K uptake by weed shoots in the apple orchard, 2019–2021.

| | Treatment | K uptake (kg ha) | | | |
|---|---|---|---|---|---|
| | | **2019** | **2020** | **2021** | **Mean** |
| 1. | No fertilization | 48.3 ± 3.19 ab | 46.0 ± 4.15 b | 43.8 ± 3.26 a | 46.0 ± 2.25 a |
| 2. | Fungal strains (fung.) to soil | 44.0 ± 3.27 a | 45.5 ± 2.97 b | 45.5 ± 3.05 ab | 45.0 ± 0.87 a |
| 3. | Bacterial strains (bact.) to soil | 51.6 ± 4.44 b | 43.4 ± 4.75 ab | 51.2 ± 2.25 c–e | 48.7 ± 4.62 ab |
| 4. | Standard NPK: urea + S. Fos Dar + potassium (K) salt | 93.2 ± 6.02 e | 77.9 ± 1.80 f | 55.9 ± 7.19 ef | 75.7 ± 18.8 d |
| 5. | Standard NPK + fung. to soil | 87.6 ± 1.91 de | 61.8 ± 1.28 d | 65.1 ± 2.00 h | 71.5 ± 14.0 cd |
| 6. | Standard NPK + bacter. to soil | 92.0 ± 7.05 e | 61.5 ± 5.06 d | 64.3 ± 4.43 h | 72.6 ± 16.9 cd |
| 7. | 100% Polifoska 6 + NK (urea + K salt) | 87.7 ± 5.54 de | 64.7 ± 4.87 d | 62.8 ± 1.17 gh | 71.7 ± 13.9 cd |

**Table 8.** *Cont.*

| | Treatment | K uptake (kg ha) | | | |
|---|---|---|---|---|---|
| | | 2019 | 2020 | 2021 | Mean |
| 8. | 100% Polifoska 6 en. bacter. + NK (urea + K salt) | 91.8 ± 2.99 e | 72.1 ± 3.51 e | 58.6 ± 3.83 fg | 74.2 ± 16.7 cd |
| 9. | 100% Urea en. fung. + PK (S. Fos Dar + K salt) | 81.2 ± 4.29 cd | 55.6 ± 4.25 c | 63.3 ± 1.44 gh | 66.7 ± 13.1 cd |
| 10. | 100% S. Fos Dar en. bact.+ NK (urea + K salt) | 80.8 ± 6.20 cd | 52.8 ± 3.02 c | 53.9 ± 0.99 d–f | 62.5 ± 15.9 b–d |
| 11. | 60% urea 60% en. fung. + 60% PK (S. Fos Dar + K salt) | 89.8 ± 7.88 e | 40.1 ± 2.29 a | 50.2 ± 0.76 b–d | 60.0 ± 26.3 a–d |
| 12. | 60% Polifoska 60% en. bacter. + 60% NK (urea + K salt) | 76.2 ± 1.73 c | 43.7 ± 1.80 ab | 54.8 ± 2.06 d–f | 58.2 ± 16.5 a–c |
| 13. | 60% S. Fos Dar 40 en. bacter. + 60% NK (urea + K salt) | 79.3 ± 5.86 c | 51.3 ± 1.61 c | 48.6 ± 2.61 a–c | 59.7 ± 17.0 a–d |

Means followed by the same letter do not differ significantly at $p$ = 0.05. Values with the prefix ± represent the standard deviation.

In 2019, weed shoots were characterized by a higher K uptake in all treatments with mineral fertilization compared to plots without such fertilization. The in-soil application of beneficial microorganisms on the plots without mineral fertilization or with standard NPK fertilization, as well as the enrichment of the full dose of Polifoska 6 + NK (urea + K salt) with bacteria, did not have a significant effect on the mean K uptake in 2019–2021, compared to the corresponding treatments without microorganisms. In 2020, the K uptake by weed shoots on plots with the NPK standard was significantly higher than in the treatments of the NPK standard + fungi or bacteria, but in 2021 the relationship between the treatments was opposite.

The mean annual Mg uptake by weed shoots in the treatment of 100% Polifoska 6 + NK (urea + K salt), amounting to 5.94 kg ha, was significantly higher than in plots without mineral fertilization, where it was 4.14–4.42 kg ha (Table 9).

**Table 9.** Mg uptake by weed shoots in the apple orchard, 2019–2021.

| | Treatment | Mg Uptake (kg ha) | | | |
|---|---|---|---|---|---|
| | | 2019 | 2020 | 2021 | Mean 2019–2021 |
| 1. | No fertilization | 3.87 ± 0.25 b | 4.54 ± 0.43 bc | 4.16 ± 0.31 a–c | 4.19 ± 0.34 ab |
| 2. | Fungal strains (fung.) to soil | 2.98 ± 0.23 a | 5.08 ± 0.30 cd | 4.37 ± 0.29 b–d | 4.14 ± 1.09 a |
| 3. | Bacterial strains (bact.) to soil | 3.92 ± 0.34 b | 4.47 ± 0.47 bc | 4.88 ± 0.22 e | 4.42 ± 0.48 a–c |
| 4. | Standard NPK: urea + S. Fos Dar + potassium (K) salt | 7.28 ± 0.42 f | 6.14 ± 0.31 ef | 4.04 ± 0.52 ab | 5.82 ± 1.64 cd |
| 5. | Standard NPK + fung. to soil | 6.03 ± 0.13 c–e | 5.05 ± 0.09 cd | 4.68 ± 0.23 de | 5.25 ± 0.70 a–d |
| 6. | Standard NPK + bacter. to soil | 5.49 ± 0.40 cd | 6.27 ± 1.00 f | 5.28 ± 0.37 f | 5.68 ± 0.52 a–d |
| 7. | 100% Polifoska 6 + NK (urea + K salt) | 6.57 ± 0.41 ef | 5.48 ± 0.42 d | 5.67 ± 0.22 g | 5.94 ± 0.57 d |
| 8. | 100% Polifoska 6 en. bacter. + NK (urea + K salt) | 6.41 ± 0.18 d–f | 6.22 ± 0.73 f | 4.63 ± 0.30 de | 5.75 ± 0.98 b–d |
| 9. | 100% Urea en. fung. + PK (S. Fos Dar + K salt) | 5.25 ± 0.28 c | 5.32 ± 0.53 d | 4.46 ± 0.10 cd | 5.01 ± 0.48 a–d |
| 10. | 100% S. Fos Dar en. bact.+ NK (urea + K salt) | 5.74 ± 0.44 c–e | 5.55 ± 0.29 de | 4.46 ± 0.08 cd | 5.25 ± 0.69 a–d |
| 11. | 60% urea 60% en. fung. + 60% PK (S. Fos Dar + K salt) | 6.00 ± 0.53 c–e | 3.68 ± 0.12 a | 3.93 ± 0.06 a | 4.54 ± 1.27 a–d |

**Table 9.** *Cont.*

| Treatment | | Mg Uptake (kg ha) | | | |
|---|---|---|---|---|---|
| | | 2019 | 2020 | 2021 | Mean 2019–2021 |
| 12. | 60% Polifoska 60% en. bacter. + 60% NK (urea + K salt) | 6.05 ± 0.14 c–e | 3.94 ± 0.16 ab | 3.83 ± 0.14 a | 4.61 ± 1.25 a–d |
| 13. | 60% S. Fos Dar 40 en. bacter. + 60% NK (urea + K salt) | 5.32 ± 0.40 c | 4.07 ± 0.13 ab | 4.11 ± 0.22 a–c | 4.50 ± 0.71 a–d |

Means followed by the same letter do not differ significantly at $p = 0.05$. Values with the prefix ± represent the standard deviation.

The greatest differences in Mg uptake between treatments were found in 2019, when weed shoots on plots of all treatments with full mineral fertilization took up significantly more Mg than weeds on plots without mineral fertilization.

The mean Ca uptake in the years 2019–2021 in NPK standard + fungi plots, which reached the value of 22.1 kg ha, was significantly higher than in plots: without mineral fertilization, 100% S. Fos Dar en. bact. + NK (urea + K salt) and 60% S. Fos Dar en. bacter. + 60% NK (urea + K salt), where it was 14.2–15.5 kg ha (Table 10).

**Table 10.** Ca uptake by weed shoots in the apple orchard, 2019–2021.

| Treatment | | Ca Uptake (kg ha) | | | |
|---|---|---|---|---|---|
| | | 2019 | 2020 | 2021 | Mean 2019–2021 |
| 1. | No fertilization | 14.0 ± 0.91 b | 16.1 ± 1.45 b | 12.6 ± 0.91 c | 14.2 ± 1.76 a |
| 2. | Fungal strains (fung.) to soil | 11.5 ± 0.82 a | 16.1 ± 1.07 b | 16.5 ± 1.11 fg | 14.7 ± 2.78 ab |
| 3. | Bacterial strains (bact.) to soil | 14.5 ± 1.24 b | 16.4 ± 1.55 b | 15.5 ± 0.70 ef | 15.5 ± 0.95 a–c |
| 4. | Standard NPK: urea + S. Fos Dar + potassium (K) salt | 22.5 ± 1.46 h | 27.2 ± 1.45 e | 12.1 ± 1.56 bc | 20.6 ± 7.73 cd |
| 5. | Standard NPK + fung. to soil | 24.4 ± 0.53 i | 26.3 ± 0.31 e | 15.7 ± 0.76 ef | 22.1 ± 5.65 d |
| 6. | Standard NPK + bacter. to soil | 20.5 ± 1.49 d–g | 23.2 ± 1.41 d | 16.5 ± 1.37 fg | 20.1 ± 3.37 b–d |
| 7. | 100% Polifoska 6 + NK (urea + K salt) | 21.2 ± 1.36 f–h | 23.0 ± 1.62 d | 15.1 ± 0.29 e | 19.8 ± 4.14 a–d |
| 8. | 100% Polifoska 6 en. bacter. + NK (urea + K salt) | 21.0 ± 0.59 e–h | 23.7 ± 1.42 d | 17.3 ± 1.13 g | 20.7 ± 3.21 cd |
| 9. | 100% Urea en. fung. + PK (S. Fos Dar + K salt) | 19.9 ± 0.94 c–f | 19.8 ± 0.82 c | 16.0 ± 0.37 ef | 18.6 ± 2.22 a–d |
| 10. | 100% S. Fos Dar en. bact.+ NK (urea + K salt) | 18.6 ± 1.43 c | 15.7 ± 0.65 b | 11.0 ± 0.19 ab | 15.1 ± 3.84 a–c |
| 11. | 60% urea 60% en. fung. + 60% PK (S. Fos Dar + K salt) | 22.0 ± 1.93 gh | 17.3 ± 0.28 b | 10.7 ± 0.17 a | 16.7 ± 5.68 a–d |
| 12. | 60% Polifoska 60% en. bacter. + 60% NK (urea + K salt) | 19.3 ± 0.46 c–e | 17.6 ± 0.92 b | 13.9 ± 0.50 d | 16.9 ± 2.76 a–d |
| 13. | 60% S. Fos Dar 40 en. bacter. + 60% NK (urea + K salt) | 19.0 ± 1.40 cd | 12.8 ± 0.34 a | 11.5 ± 0.64 a–c | 14.4 ± 4.01 a |

Means followed by the same letter do not differ significantly at $p = 0.05$. Values with the prefix ± represent the standard deviation.

In 2019, the uptake of Ca by weed shoots in all plots with mineral fertilization was significantly higher than in plots where such fertilization were not applied. In 2021, the uptake of Ca in plots treated with beneficial organisms was significantly higher compared to plots where they were not used, and it concerned both control plots (no treatment), NPK standard and 100% Polifoska 6 + NK (urea + K salt).

### 3.6. The Content of Macronutrients in Apple Leaves

Apple leaves from the experimental plots, in relation to the compositional standards, were characterized by a high average content of N, P, K and a low content of Mg and Ca (Table 11).

**Table 11.** Mean macronutrient content in apple leaves, 2019–2021.

| | Treatment | Macronutrient Content (% DW) | | | | |
|---|---|---|---|---|---|---|
| | | N | P | K | Mg | Ca |
| 1. | No fertilization | 2.51 ± 0.15 a–c | 0.20 ± 0.04 c | 1.36 ± 0.18 a–d | 0.21 ± 0.01 d | 1.54 ± 0.11 d |
| 2. | Fungal strains (fung.) to soil | 2.47 ± 0.18 a | 0.19 ± 0.06 bc | 1.32 ± 0.24 a–d | 0.19 ± 0.03 a–d | 1.51 ± 0.20 a–d |
| 3. | Bacterial strains (bact.) to soil | 2.48 ± 0.19 ab | 0.17 ± 0.02 ab | 1.19 ± 0.24 a | 0.20 ± 0.04 cd | 1.51 ± 0.20 b–d |
| 4. | Standard NPK: urea + S. Fos Dar + potassium (K) salt | 2.69 ± 0.14 a–d | 0.17 ± 0.01 ab | 1.47 ± 0.19 d | 0.18 ± 0.03 a–c | 1.53 ± 0.14 cd |
| 5. | Standard NPK + fung. to soil | 2.71 ± 0.20 d | 0.15 ± 0.01 a | 1.27 ± 0.34 a–d | 0.18 ± 0.04 a–c | 1.46 ± 0.25 a–d |
| 6. | Standard NPK + bacter. to soil | 2.64 ± 0.26 a–d | 0.15 ± 0.02 a | 1.26 ± 0.36 a–c | 0.17 ± 0.05 a–c | 1.47 ± 0.19 a–d |
| 7. | 100% Polifoska 6 + NK (urea + K salt) | 2.69 ± 0.19 b–d | 0.15 ± 0.01 a | 1.41 ± 0.28 cd | 0.17 ± 0.03 ab | 1.50 ± 0.19 a–d |
| 8. | 100% Polifoska 6 en. bacter. + NK (urea + K salt) | 2.67 ± 0.22 a–d | 0.15 ± 0.02 a | 1.36 ± 0.41 a–d | 0.18 ± 0.02 a–c | 1.43 ± 0.09 ab |
| 9. | 100% Urea en. fung. + PK (S. Fos Dar + K salt) | 2.68 ± 0.19 a–d | 0.15 ± 0.01 a | 1.39 ± 0.29 b–d | 0.17 ± 0.03 ab | 1.42 ± 0.18 a |
| 10. | 100% S. Fos Dar en. bact.+ NK (urea + K salt) | 2.69 ± 0.19 cd | 0.15 ± 0.01 a | 1.25 ± 0.27 ab | 0.19 ± 0.04 a–d | 1.49 ± 0.21 a–d |
| 11. | 60% urea 60% en. fung. + 60% PK (S. Fos Dar + K salt) | 2.62 ± 0.08 a–d | 0.15 ± 0.01 a | 1.37 ± 0.26 a–d | 0.17 ± 0.03 ab | 1.48 ± 0.21 a–d |
| 12. | 60% Polifoska 60% en. bacter. + 60% NK (urea + K salt) | 2.64 ± 0.10 a–d | 0.15 ± 0.02 a | 1.34 ± 0.38 a–d | 0.17 ± 0.03 a–c | 1.48 ± 0.12 a–d |
| 13. | 60% S. Fos Dar 40 en. bacter. + 60% NK (urea + K salt) | 2.64 ± 0.20 a–d | 0.15 ± 0.01 a | 1.29 ± 0.28 a–d | 0.19 ± 0.05 b–d | 1.45 ± 0.19 a–c |
| | Optimal range | 2.10–2.40 * | 0.15–0.26 * | 1.0–1.50 * | 0.22–0.32 * | 2.1–2.5 ** |

Means followed by the same letter do not differ significantly at $p = 0.05$. Values with the prefix ± represent the standard deviation. * Nutrient ranges recommended for apples in Poland [76]. ** Nutrient ranges recommended for apples in Australia [77].

The mean N content ranged from 2.47 (fungal strains) to 2.71% DW (NPK standard + fungal strains). Apple leaves from standard NPK + fungal strains plots contained significantly more N than leaves from plots without mineral fertilization. The mean P content in apple leaves in the untreated plots, amounting to 0.2% DW, was significantly higher than in the plots where bacterial strains and mineral fertilizers were applied, at a full or reduced dose (0.15% DW). The mean K content in apple leaves fertilized with the NPK standard (1.47% DW) was significantly higher than in the plots treated with bacterial strains (1.19% DW), NPK standard + bacteria strains (1.27% DW), and 100% S. Fos Dar en. bact. + NK (urea + K salt) (1.25% DW). Apple leaves growing on untreated plots had a significantly higher Mg content (0.21% DW) than leaves from plots where mineral fertilizers were used (0.17–0.18 DW), except for two treatments—100% S. Fos Dar en. bact. + NK (urea + K salt) and 60% S. Fos Dar 40 en. bacter. + 60% NK (urea + K salt). Significant differences in Ca content were found only between apple leaves growing on untreated plots (1.54% DW), where it was higher than in the treatments of 100% Polifoska 6 en. bacter. + NK (urea + K salt), 100% urea en. fung. + PK (S. Fos Dar + K salt), 60% S. Fos Dar en. bacter. + 60% NK (urea + K salt) (1.42–1.45% DW).

### 3.7. The content of Macronutrients in the Orchard Soil

The apple orchard soil in the plots of all treatments, in the last year of the research, was characterized by a high content of P (10.6–13.4 mg 100 g of soil), K (11.0–22.2 mg 100 g of soil), Mg (5.87–7.77 mg 100 g of soil)) in relation to the standards (Table 12).

**Table 12.** Macronutrient content in apple orchard soil, 2021.

| | Treatment | Macronutrient Content (mg 100 g Soil) | | | K: Mg Ratio |
|---|---|---|---|---|---|
| | | **P** | **K** | **Mg** | |
| 1. | No fertilization | 11.7 ± 0.17 a–c | 13.3 ± 0.98 ab | 6.74 ± 0.11 a–c | 1.97 |
| 2. | Fungal strains (fung.) to soil | 12.0 ± 0.52 a–c | 11.0 ± 1.15 a | 7.35 ± 1.32 bc | 1.50 |
| 3. | Bacterial strains (bact.) to soil | 10.6 ± 0.46 a | 12.2 ± 0.23 ab | 6.71 ± 0.01 a–c | 1.81 |
| 4. | Standard NPK: urea + S. Fos Dar + potassium (K) salt | 13.3 ± 1.39 c | 22.2 ± 2.71 f | 5.87 ± 0.42 a | 3.78 |
| 5. | Standard NPK + fung. to soil | 10.8 ± 0.92 ab | 14.6 ± 1.67 a–c | 6.96 ± 1.07 a–c | 2.10 |
| 6. | Standard NPK + bacter. to soil | 13.4 ± 1.33 c | 20.4 ± 1.94 ef | 7.77 ± 0.79 c | 2.63 |
| 7. | 100% Polifoska 6 + NK (urea + K salt) | 13.3 ± 0.17 c | 18.2 ± 0.06 c–e | 7.26 ± 0.07 bc | 2.51 |
| 8. | 100% Polifoska 6 en. bacter. + NK (urea + K salt) | 12.9 ± 0.29 c | 19.1 ± 3.06 d–f | 7.13 ± 0.57 bc | 2.68 |
| 9. | 100% Urea en. fung. + PK (S. Fos Dar + K salt) | 13.3 ± 1.27 c | 19.5 ± 2.42 d–f | 6.76 ± 0.61 a–c | 2.88 |
| 10. | 100% S. Fos Dar en. bact.+ NK (urea + K salt) | 11.4 ± 1.74 a–c | 17.2 ± 0.58 c–e | 6.52 ± 0.92 ab | 2.64 |
| 11. | 60% urea 60% en. fung. + 60% PK (S. Fos Dar + K salt) | 12.8 ± 2.30 bc | 15.7 ± 0.52 b–d | 6.78 ± 0.03 a–c | 2.32 |
| 12. | 60% Polifoska 60% en. bacter. + 60% NK (urea + K salt) | 11.7 ± 1.21 a–c | 17.4 ± 1.91 c–e | 6.94 ± 0.11 a–c | 2.51 |
| 13. | 60% S. Fos Dar 40 en. bacter. + 60% NK (urea + K salt) | 12.9 ± 1.85 c | 12.9 ± 1.85 ab | 7.35 ± 0.40 bc | 1.76 |
| | Average range * | 2–4 | 5–8 | 2.5–4 | |
| | High range | >4 | >8 | >4 | |

Means followed by the same letter do not differ significantly at $p = 0.05$. Values with the prefix ± represent the standard deviation. * Nutrient ranges recommended for apples in Poland [76].

The soil from plots treated with bacterial strains contained significantly less P (10.6 mg 100 g of soil) than in plots with mineral fertilization (12.8–13.4 mg 100 g of soil), with the exception of three treatments: standard NPK, 100% S. Fos Dar en. bact. + NK (urea + K salt), and 60% Polifoska 6 en. bacter. + 60% NK (urea + K salt). K content in the soil on the plots of the treatments: 60% Polifoska 6 en. bacter. + 60% NK (urea + K salt) and fertilized with a full dose of mineral fertilizers, except for standard NPK + fungal strains, reached values 17.2–22.2 mg 100 g of soil and was significantly higher than on plots without mineral fertilization (11.0–13.3 mg 100 g of soil). In the soil of plots with standard NPK fertilization, mixed with bacterial strains, a significantly higher content of Mg (7.77 mg 100 g of soil) was found than in plots with NPK standard (5.87 mg 100 g of soil). The standard NPK treatment was the only one, found to have an incorrect K: Mg ratio of greater than 3.5 (Table 12).

### 3.8. Correlation between Fertilization, Macronutrient Uptake of Weeds and Nutritional Status of Apple Trees

A very strong or strong positive correlation was found between the fresh weight of weeds; uptake of N, P, K Mg, Ca by weeds; and the level of NPK mineral fertilization (Table 13). However, no such correlation was found between the fresh weight of weeds, the uptake of macronutrients by weeds, the content of macronutrients in apple leaves and the use of beneficial microorganisms. A significant positive correlation between mineral

fertilization and the content of macronutrients in apple leaves was demonstrated only for N. The content of P, Mg and Ca in apple leaves was negatively correlated with the uptake of these macronutrients by weeds, and the strength of the correlation was weak to moderate.

**Table 13.** Selected correlation coefficients between NPK fertilization level, beneficial microorganism level, fresh weight of weeds, macroelements uptake by weeds and macroelements content in apple leaves and orchard soil.

| Trait | NPK Fertilization Level | Beneficial Microorganisms Level | Uptake of Macronutrients by Weed Shoots | | | | |
|---|---|---|---|---|---|---|---|
| | | | N | P | K | Mg | Ca |
| | | | Coefficient *r* | | | | |
| Fresh weight of weed shoots | 0.929 | −0.099 | 0.987 | 0.933 | 0.989 | 0.884 | 0.883 |
| N uptake by weed shoots (s.) | 0.953 | −0.009 | - | - | - | - | - |
| P uptake by weeds s. | 0.843 | −0.086 | - | - | - | - | - |
| K uptake by weeds s. | 0.948 | −0.103 | - | - | - | - | - |
| Mg uptake by weeds s. | 0.862 | −0.272 | - | - | - | - | - |
| Ca uptake by weeds s. | 0.748 | −0.113 | - | - | - | - | - |
| N content in apple leaves | 0.954 | 0.039 | 0.912 | - | - | - | - |
| P content in apple leaves | −0.784 | −0.446 | - | −0.577 | - | - | - |
| K content in apple leaves | 0.294 | −0.631 | - | - | 0.358 | - | - |
| Mg content in apple leaves | −0.705 | −0.191 | - | - | - | −0.543 | - |
| Ca content in apple leaves | −0.536 | −0.644 | - | - | - | - | −0.346 |
| P content in orchard soil | 0.490 | −0.258 | - | 0.647 | - | - | - |
| K content in orchard soil | 0.809 | −0.241 | - | - | 0.831 | - | - |
| Mg content in orchard soil | −0.045 | 0.377 | - | - | - | −0.044 | - |

Strength of Pearson's linear correlation coefficient |*r*|: 0.0–0.2—very weak; 0.2–0.4—weak; 0.4–0.6—moderate; 0.6–0.8—strong; 0.8–1—very strong.

## 4. Discussion

The species composition of weed communities on all plots was similar, which is consistent with the statement that tree-row management has a stronger impact on the weed community than the fertilization strategies [78]. Weed species, which were found in the present research, include *Poa annua*, *Stellaria media*, *Capsella bursa-pastoris*, *Senecio vulgaris*, *Chenopodium album*, *Echinochloa crus-galli*, *Amaranthus retroflexus* and *Lamium amplexicaule*, which commonly occur in weed communities in apple orchards grown in various cultivation systems, as well as in other fruit crops, in Eurasian countries, ranging from Western Europe to the Far East [79–84]. Weed infestation in orchards can be expressed as soil cover by plants of particular weed species [79] or as their number per unit area [81]. In the present research, a better indicator was soil coverage with weeds, as fertilization primarily modifies the growth strength of weeds [15]. The soil coverage with weeds in all plots with mineral fertilization, regardless of its dose or the addition of beneficial microorganisms, was significantly greater than in the plots not treated or treated only with beneficial organisms—fungal or bacterial strains. The response of weeds to mineral fertilization was varied in line with the preferences for soil fertility described in the literature. The neutrophilic species that reacted positively to N fertilization were AMARE [15,21], CAPBP [15,22], SENVU [15,20], and STEME [15,18]. The species that covered the soil to a greater extent in plots with mineral fertilization than in non-fertilized plots were CHEAL and ECHCG. This confirmed reports that both CHEAL [15,16] and ECHCG [15,19] grow strongly on fertile soils. The reaction of POAAN, which is tolerant to soil conditions [17], was relatively weaker, although it develops better on nitrogen-fertilized soils [15]. The soil coverage by LAMAM was not dependent on the fertilization method, as it is a plant with no clear preferences as to soil fertility [15].

The fresh weight of the weed shoots is more precise than the soil coverage, and at the same time an easily measurable parameter that indicates the effect of fertilization on the growth of weeds. Mean seasonal fresh weight of weeds from three years was significantly higher for six out of seven treatments with full mineral fertilization than for three treatments without mineral fertilization. The mutual relations between the treatments, however, differed from season to season. In 2019, the fresh weight of weeds on plots

without mineral fertilization was significantly lower than in all other plots where mineral fertilization was applied, with or without fungal or bacterial strains. In 2021, the fresh weight of weeds was significantly lower in the untreated plots than in all other treatments, including those where only beneficial microorganisms were used. This may indicate that plants used macronutrients from the decomposition and mineralization of weed residues controlled in previous seasons for their growth, which is observed both in orchards [29] and in agricultural crops such as rice [32]. The soil organisms used, according to reports [66,68], could improve the process of macronutrient uptake by plants.

The method of application, which was varied under the conditions of the experiment, is of significant importance for the activity of beneficial microorganisms [56,57]. In treatments where consortia of fungi or bacteria were mixed with the soil (treatments labeled 5 and 6), there was a trend towards an increase in fresh weight of weeds and N, P, K and Ca uptake, compared to their counterparts with the same mineral fertilization, where the fungi or the bacteria were added to one of the mineral fertilizers (treatments 9 and 10).

The differences in the mean content of air-DW of the weed shoots were small and statistically significant only between the NPK standard (the lowest value) and the 60% Polifoska 6 en. bacter. + 60% NK (urea + K salt) treatment. The use of fungal or bacterial consortia on the plots without mineral fertilization, with the NPK standard, and the bacterial consortium with the 100% Polifoska 6 + NK (urea + K salt) facility did not change the mean content of all nutrients in the weed shoots and the mean N, P, K, Mg and Ca uptake by weeds, compared to the corresponding treatments, where no microorganisms were used. The N and P content in weed shoots from the present research was consistent with the data relating to weeds from Polish orchards [10] and New Zealand pastures [85], and higher than in weeds in Bulgarian tobacco [86] and South Korean orchards [29]. The content of K and Mg was similar or higher, depending on the cited author, while the content of Ca was similar to that of all other authors [10,29,85,86]. The nutrients uptake by weeds is the most important indicator of nutrient competition between weeds and crop [86]. The mean annual uptake of N in the period May–August in the presented studies was similar, and P and K were significantly lower compared to pear orchards in South Korea [30]. On the other hand, the uptake of all macronutrients was significantly higher than that found in tobacco plantations [86]. The reason for the significant variability in the content of macronutrients in weeds and their uptake by weeds are many factors, the most important of which are the species composition of weeds [10,85], the ability of weeds to compete with the cultivated plant [13,14,26,27], agricultural practices related to fertilization and weed control as well as climate and soil conditions [49,56,87].

Interpretation of the relationship between fertilization, macronutrient losses and the nutritional status of crop plants was facilitated by Pearson's linear correlation coefficient, which was calculated for some pairs of parameters. As expected and logically, there was a strong positive correlation between the macronutrient uptake by weeds and the fresh weight of the weed shoots. A very strong or strong positive correlation was found between the uptake of macronutrients by weeds and the level of mineral fertilization. Mineral fertilization had a significant effect only on the N nutritional status of apple trees, expressed as the content of N in apple leaves. The correlation coefficient between the fertilization level and the macronutrient content in apple leaves was low for K and negative for P. Thus, it can be concluded that the benefits of mineral fertilization with NPK, also with the addition of beneficial organisms, are in the case of N with both weeds and apple trees, while in the case of P and K most of all weeds. A similar pattern was found for wheat, where PK fertilization favored weeds [88]. However, no clear correlation was found between the use of beneficial microorganisms and the uptake of nutrients by weeds and the content of macronutrients in apple leaves. The nutritional status of apple trees in the present studies did not depend on the method of fertilization. The mean content of N in apple leaves in the plots of all treatments was excessively high, P and K were optimal, and Mg and Ca were low compared to the reference limits [76,77].

The results from the present studies, referring to a different period (2018–2020) and limited to the first six treatments, showed that in some years beneficial microorganisms can increase the N content in apple leaves, both on plots without mineral fertilization and those with mineral fertilization, and the P and K content mainly on non-fertilized plots [89]. The present research did not confirm the results of the experiment carried out on strawberry plantations, where the same beneficial microorganisms increased the content of P and Ca in strawberry leaves [71], nor the results of field experiments in apple trees, where *Bacillus* bacteria increased the content of macronutrients in apple leaves [63,67]. The effect of using beneficial organisms such as four PGPR strains (*Bacillus* OSU-142, *Bacillus* M-3, *Burkholderia* OSU-7 and *Pseudomonas* BA-8) on apple yield was strongly dependent on rootstocks, cultivars and treatments [90]. It should be expected that the above-mentioned factors may also be important for the mineral status of apple trees.

The availability of nutrients for a crop, usually improved by beneficial microorganisms, could be modified by weed–crop competition [13,14], as well as the interaction of microorganisms and synanthropic flora [42,91]. The results of the present research confirmed that with the increase in the N content in apple leaves, the P content decreases [92]. The results concerning the content of macronutrients in tree leaves should be interpreted with reserve, as the tendency to lower the content of N, Mg and Ca may be caused by a strong growth of trees [93]. The obtained results conclude that in fruit-bearing orchards with good availability of macronutrients in the soil, the periodic presence of strongly growing weeds and high uptake of macronutrients by weeds do not constitute a significant nutrient competition for fruit trees, as reported by other authors [94,95].

In the present experiment, weed control three times in an apple orchard in the period April–August maintained the mineral nutritional status of trees, regardless of the doses of mineral fertilization and the use of beneficial microorganisms, consortia of fungi or bacteria, applied alone or together with mineral fertilizers. However, it should be taken into account, that in practice, weeds not only compete for nutrients, but also pose other serious threats to trees, such as rodents [10]. Therefore, weeds should be systematically controlled to ensure good tree growth and high yield and fruit quality [4,7,8]. On the other hand, taking into account the possibility of practical implementation of modern OFM and the role of weeds as a provider of environmental services, it is necessary to develop an economically justified model of the periodic reduction of weed infestation, which will reduce the threats caused by weeds and will preserve the environmental benefits of weeds.

## 5. Conclusions

The weight of weeds and the uptake of macronutrients by weeds in the apple orchard increased significantly more after the application of mineral fertilization than after the application of beneficial microorganisms—fungi and bacteria consortia. The use of beneficial microorganisms on the plots without mineral fertilization and together with NPK fertilization did not change the mean nutrient uptake by weeds over the three-year period, compared to the appropriate treatments where no microorganisms were used. In some years, the use of microorganisms increased macronutrient uptake by weeds compared to untreated control or NPK fertilization, which proves that the microorganisms improve the availability of macronutrients to weed plants. The increase in weed weight and the uptake of N, P, K and Ca by weeds was slightly more visible when the microorganisms were incorporated into the soil than when they were spread over the soil surface after mixing with mineral fertilizers. In a fruit-bearing apple orchard growing on a soil with good fertility, with weeding three times in the period April–August, the increased uptake of macronutrients by weeds did not significantly change the nutritional status of apple trees expressed as the content of N, P, K Mg and Ca in apple leaves. There was only a weak to moderate tendency to reduce the content of P, Mg and Ca in apple leaves with an increase in the uptake of these macronutrients by weeds. After using NPK mineral fertilization, also with the addition of beneficial organisms, the uptake of N, both by weeds and apple trees

increased. P and K were more efficiently absorbed by weeds and they achieved greater benefits from fertilization with these two macronutrients than trees.

**Author Contributions:** Conceptualization, J.L. and L.S.-P.; methodology, J.L. and L.S.-P.; formal analysis, J.L.; investigation, A.M. and J.L.; data curation, J.L.; writing—original draft preparation, J.L.; writing—review and editing, A.L. and L.S.-P.; supervision, L.S.-P.; funding acquisition, L.S.-P. All authors have read and agreed to the published version of the manuscript.

**Funding:** This publication was financed (co-financed) by the National Center for Research and Development under the BIOSTRATEG program, contract number BIOSTRATEG3/347464/5/NCBR/2017.

**Data Availability Statement:** Data are contained within this article.

**Conflicts of Interest:** The authors declare no conflict of interest.

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
