# Peer review of "The Response of Weeds and Apple Trees to Beneficial Soil Microorganisms and Mineral Fertilizers Applied in Orchards"

_agronomy, doi:10.3390/agronomy12112882_

Round 1

Reviewer 1 Report

Manuscript: agronomy-1997928

Version: peer-review-v1

>> The theme of the manuscript is integrated nutrient management in apple orchards in Poland. The Topic of research highlighted in manuscript is of considerable interest.

>> However, the quality of presentation and text is not upto the mark. There were several issue with the manuscript which must need significant rewriting of the manuscript.

>> Due to the amount of errors/issue, commenting as point to point is not possible for the current version.

>> Some points around the issues are:

* Introduction was too long and too much of extra/unnecessary information. Some information can be used in discussion where it would fir better.

* Too many too long sentences. These sentence are maybe too complicated that authors might have not read them again as their meaning sometimes is not so logical.

* I could not understand the usage of units in the manuscript.

* Citations after 30 are shifted and thus incorrect.

* The font size which was small and larger font size was annoying.

* Too many and many times too irrelevant citations.

Author Response

Skierniewice, 08.11.2022

Response to the Reviewer's 1 comments

The authors wish to thank the Reviewer for your valuable comments that will improve the quality of the reviewed manuscript "The response of weeds and apple trees to beneficial soil microorganisms and mineral fertilizers applied in orchard” (agronomy-1997928). We have completed the manuscript  according to the Reviewers comments. The style has been improved.  The text has been corrected or supplemented to take into account the comments of the Reviewer.

A list of the corrections made, recommended by the Reviewer:

Ad. >> However, the quality of presentation and text is not up to the mark. There were several issue with the manuscript which must need significant rewriting of the manuscript<<.

Many parts of the manuscript was rewriting.

Ad. >> Introduction was too long and too much of extra/unnecessary information. Some information can be used in discussion where it would fir better<<.

The introduction has been shortened by more than one page. Part of the information was used in the discussion.

Ad. >> Too many too long sentences. These sentence are maybe too complicated that authors might have not read them again as their meaning sometimes is not so logical.

Many sentences have been shortened (split) or re-edited.

Ad. >> I could not understand the usage of units in the manuscript.<<

Units have been improved.

Ad. >> Citations after 30 are shifted and thus incorrect.<< and << The font size which was small and larger font size was annoying.>>

The manuscript was submitted as an original MS Word file. In the Editorial Office it was transformed into an MDPI-template. Errors occurred while converting the file. They concerned different font sizes in particular parts of the document and references list,  where the citation sequence numbers were duplicated in two places, which resulted in incorrect citations in the text. The authors received the MDPI-template version only with the reviews and had no influence on the quality of the text sent to the reviewers. Of course, we understand the Reviewer's comments and dissatisfaction. We kindly inform you, that we have never consciously sent such a poorly edited manuscript.

Ad. <<Too many and many times too irrelevant citations>>

In the introduction and discussion, many citations were indeed used. Information was needed on weeds (biology, harmfulness and control), fertilization, weed-crop competition, apple cultivation and beneficial microorganisms (in general and about microorganisms used in the experiment). The intention of the authors was to draw attention to the complexity of the interactions between the categories of microorganisms - weeds - crop. The literature data mainly concern the microorganism-crop or weed-crop system. Relationships about the weed (rather flora) - microorganism system are very few. The system of microorganisms - weeds - cultivated plant is practically not a subject of research, especially in terms of fertilization. As recommended, some citations have been moved from the introduction to the discussion. Some citation have been abandoned. 

Yours Sincerely

Jerzy Lisek

Reviewer 2 Report

The manuscript describes the response of weeds and apple trees to beneficial soil microorganisms and mineral fertilizers applied in orchard.

 The text is well-written. Some points are raised below. All answers should be included in the text.

 1.           The introduction section is too extended; it should be at least one page shorter.

2.           “…the soil pH was slightly acidic at pH 6.2 (in KCl)…”: not clear, please, explain.

3.           Why did the authors choose apple trees for their study? Some explanation is needed.

4.           Further explanation is also needed regarding the changes observed over the years 2019, 2020, 2021 for all the parameters examined:

a)       Coverage of the soil with weeds

b)      Fresh weight of weed shoots

c)       The content of air-DW of weed shoots

d)      Macronutrients content in weed shoots

e)      Uptake of macronutrients by weed shoots

f)        The content of macronutrients in apple leaves

g)       The content of macronutrients in the orchard soil

What conclusions can be derived from these fluctuations?

Author Response

Skierniewice, 08.11.2022

Response to the Reviewer's 2 comments

The authors wish to thank the Reviewer for your valuable comments that will improve the quality of the reviewed manuscript "The response of weeds and apple trees to beneficial soil microorganisms and mineral fertilizers applied in orchard” (agronomy-1997928). We have completed the manuscript  according to the Reviewers comments. The style has been improved.  The text has been corrected or supplemented to take into account the comments of the Reviewer.

A list of the corrections made, recommended by the Reviewer:

Ad. 1.           <<The introduction section is too extended; it should be at least one page shorter.>>

The introduction has been shortened by more than one page

Ad. 2.          << “…the soil pH was slightly acidic at pH 6.2 (in KCl)…”: not clear, please, explain.>>

the text in the section Materials and Methods has been changed to the following: “At the time of planting, the pH of soil was 6.2 (in KCl)….”

Ad. 3.           <<Why did the authors choose apple trees for their study? Some explanation is needed.>>

An explanation has been added at the end of the Introduction section: “The apple tree was selected for the study because it is the most important fruit crop in the temperate climate zone.”

Ad. 4.           <<Further explanation is also needed regarding the changes observed over the years 2019, 2020, 2021 for all the parameters examined:

  1. a)      Coverage of the soil with weeds
  2. b)      Fresh weight of weed shoots
  3. c)      The content of air-DW of weed shoots
  4. d)      Macronutrients content in weed shoots
  5. e)      Uptake of macronutrients by weed shoots
  6. f)      The content of macronutrients in apple leaves
  7. g)     The content of macronutrients in the orchard soil

What conclusions can be derived from these fluctuations?>>

Data on: uptake of macronutrients by weed shoots and fresh weight of weeds is presented year by year and as a three-year average and used for discussion. As for the remaining parameters, only some of the results are presented in the article. The data obtained, which were not helpful in the interpretation of the results, were not shown. The main reason for this decision is excessive manuscript volume. In the revised manuscript, a few sentences have been added to the beginning of the result section: “Results for parameters such as: soil coverage with weeds, air-DW content of weeds, macronutrient content in apple leaves are only represented as three-year means. Data from individual years would not change the interpretation of the results. The variability of the above-mentioned parameters can be concluded on the basis of the standard deviation available in the appropriate tables.”  

Yours Sincerely

Jerzy Lisek

Author Response

Skierniewice, 08.11.2022

Response to the Reviewer's 3 comments

The authors wish to thank the Reviewer for your valuable comments that will improve the quality of the reviewed manuscript "The response of weeds and apple trees to beneficial soil microorganisms and mineral fertilizers applied in orchard” (agronomy-1997928). We have completed the manuscript  according to the Reviewers comments. The style has been improved.  The text has been corrected or supplemented to take into account the comments of the Reviewer.

A list of the corrections made, recommended by the Reviewer:

Ad. <<1). Moreover, reading the abstract, it is not very clear how many and which treatments are studied, so the authors should specify all treatments tested;>>

A sentence has been added to the abstract: “In the field experiment, the effect of thirteen treatments was investigated, where fungal or bacterial inocula and mineral fertilizers at a standard dose and a dose reduced by 40% were applied individually or together.”

 Ad. <<2). The experimental design must be restructured for the information be more clearly understood; as suggestions - a table with treatments used and a code for each treatment;>>

In the first draft manuscript there was an additional table with data on the treatments and the codes for each treatment. It was abandoned because the text was too large (additional table). As for the codes, I find that the verbal description of treatments makes it easier to track results. The description of the treatments provided in the section 2.2. Experimental design is very detailed and should not raise any doubts. At the end of section 2.2, a sentence was added: “The numbering of the treatments (1-13) used in this subsection is kept in Tables 2-11 containing the results.“ In fact, the tables contain a complete list of treatments.

Ad. <<3). Lines 181, 186 - need to correct ‘microrganisms’ with microorganisms.

4). Lines 221, 222- need to correct – m-1 , respectively m-2 with m and m2

5). Lines 225, 228, 229, 230, 233, 234, 240, 241, 244, 246, 248, 249, 251, 252 need to correct – g -1 with g;

6) Line 255 - need to correct – kg-1 with kg

7) Line 256 - need to correct – kg-1 with kg, and per ha-1 with per ha.>>

All recommended corrections have been made.

 Ad. <<8). The citations in that article is inaccurate: the numerical information reported in the article differs from that given in the source article. The References section contains errors in cited sources in the scientific paper:>>

The citation numbering was wrong and has been corrected. 

Yours Sincerely

Jerzy Lisek

Round 2

Reviewer 3 Report

The manuscript has been sufficiently improved to warrant publication in Agronomy.